# Body Mass Index is Strongly Associated with Hypertension: Results from the Longevity Check-Up 7+ Study

**DOI:** 10.3390/nu10121976

**Published:** 2018-12-13

**Authors:** Francesco Landi, Riccardo Calvani, Anna Picca, Matteo Tosato, Anna Maria Martone, Elena Ortolani, Alex Sisto, Emanuela D’Angelo, Elisabetta Serafini, Giovambattista Desideri, Maria Tecla Fuga, Emanuele Marzetti

**Affiliations:** 1Fondazione Policlinico Universitario “Agostino Gemelli”, Università Cattolica del Sacro Cuore, L.go F. Vito 8, 00168 Rome, Italy; riccardo.calvani@gmail.com (R.C.); anna.picca1@gmail.com (A.P.); matteo.tosato@policlinicogemelli.it (M.T.); annamariamartone@gmail.com (A.M.M.); eleort@gmail.com (E.O.); alexsisto@gmail.com (A.S.); manud1983@yahoo.it (E.D.); elisabettaserafini.es@gmail.com (E.S.); emanuele.marzetti@policlinicogemelli.it (E.M.); 2Department of Life, Health and Environmental Sciences, Università dell’Aquila, Via G. Petrini, Edificio Delta 6, 67100 Coppito (AQ), Italy; giovambattista.desideri@cc.univaq.it (G.D.); mariateclafuga@virgilio.it (M.T.F.)

**Keywords:** body mass index, obesity, hypertension

## Abstract

The present study was undertaken to provide a better insight into the relationship between different levels of body mass index (BMI) and changing risk for hypertension, using an unselected sample of participants assessed during the Longevity Check-up 7+ (Lookup 7+) project. Lookup 7+ is an ongoing cross-sectional survey started in June 2015 and conducted in unconventional settings (i.e., exhibitions, malls, and health promotion campaigns) across Italy. Candidate participants are eligible for enrolment if they are at least 18 years of age and provide written informed consent. Specific health metrics are assessed through a brief questionnaire and direct measurement of standing height, body weight, blood glucose, total blood cholesterol, and blood pressure. The present analyses were conducted in 7907 community-living adults. According to the BMI cutoffs recommended by the World Health Organization, overweight status was observed among 2896 (38%) participants; the obesity status was identified in 1135 participants (15%), with 893 (11.8%) participants in class I, 186 (2.5%) in class II, and 56 (0.7%) in class III. Among enrollees with a normal BMI, the prevalence of hypertension was 45% compared with 67% among overweight participants, 79% in obesity class I and II, and up to 87% among participants with obesity class III (*p* for trend < 0.001). After adjusting for age, significantly different distributions of systolic and diastolic blood pressure across BMI levels were consistent. Overall, the average systolic blood pressure and diastolic blood pressure increased significantly and linearly across BMI levels. In conclusion, we found a gradient of increasing blood pressure with higher levels of BMI. The fact that this gradient is present even in the fully adjusted analyses suggests that BMI may cause a direct effect on blood pressure, independent of other clinical risk factors.

## 1. Introduction

Despite health policies implemented in the last decades and increasingly available therapeutic options, hypertension and cardiovascular disease (CVD) continue to be the main cause of disability, morbidity and mortality, both in Europe and in the United States [1,2]. Primary and secondary prevention continues to address individuals who have experienced disease or present with one or more risk factors; furthermore, primordial prevention has been suggested for the attainment of global health at the population level [3,4]. In fact, the World Health Organization (WHO) recently highlighted the importance of controlling blood pressure as millions of individuals have CVD worldwide, accounting for 31% of the world’s population, of which 80% experience either heart diseases and stroke. Additionally, WHO dealt with high systolic and/or diastolic blood pressure as a severe problem and considered obesity as one of the most important risk factors to address (http://www.who.int/mediacentre/news/releases/2012).

Previous studies have documented the main factors of hypertension including age, gender, smoking, exercise, family history, dietary habits, and body mass index (BMI) [5,6,7]. The prevalence of obesity has been increasing throughout the world, including Europe [8,9]. It is a decisive risk factor in many chronic diseases such as hypertension, dyslipidemia, and diabetes mellitus type 2 [10,11,12]. High body weight and obesity, by means of BMI measure, are the main causes of these disorders; hence, with the importance on continuous weight management, research is actively being conducted [13].

A systematic review demonstrated that meeting higher numbers of ideal health metrics, including a normal BMI, was correlated with lower prevalence and incidence of both cardiovascular and non-cardiovascular diseases [14]. The link between obesity and hypertension is complex, considering that obesity-related hypertension is closely associated with other diseases in the course of the obesity. In general, obesity, which is usually determined by BMI, is one the principal risk factors for hypertension [8,15], and the prevalence of hypertension increases with rising BMI [16,17]. However, BMI, as the most frequent anthropometric measure used, does not reflect body fat distribution, and there has recently been some doubt concerning it as a convenient indicator of high body weight and obesity. Similarly, there are concerns about its capability to predict the risk of hypertension and CVD [18,19].

The present study was, therefore, undertaken to provide a better insight into the linear relationship between different levels of BMI and changing risk for hypertension, using an unselected sample of participants assessed during the Longevity Check-up 7+ (Lookup 7+) project.

## 2. Materials and Methods

For the present study, the database Lookup 7+ is used. The Lookup 7+ study protocol has been described in detail elsewhere [20,21]. Briefly, this project is an ongoing initiative developed by the Geriatric Medicine Department of the Università Cattolica del Sacro Cuore of Rome planned with the aim to improve healthy lifestyles in the general population [22]. People entering public environments (i.e., exhibitions, shopping centers, and sport events) or those participating in prevention campaigns have been screened using a specific questionnaire on lifestyle and had a brief check-up, specifically assembled on the basis of the ideal cardiovascular health metrics indicated by the American Heart Association [23]. Participants are selected as eligible for the check-up if they are at least 18 years of age and provide written informed consent. The only exclusion criteria are: self-reported pregnancy, incapacity to perform the physical performance tests, denial of blood capillary check, and the inability or the refusal to give written informed consent. The Università Cattolica del Sacro Cuore Ethical Committee ratified the study protocol [24].

### 2.1. Study Sample

Between June 1st, 2015 and October 30th, 2017, we enrolled 8040 participants. Recruitment took place in several Italian cities adhering to the Lookup 7+ project: Milan EXPO 2015 (Milan, June–October 2015), Mese del Cuore (Rome, September–October 2016), La Romanina—Check your Longevity (Rome, December 2017), Mese del Cuore (Milan, March-April 2017), Ministry of Health—Women’s Day (Rome, April 2017), CamBio Vita (Catania, May 2017), COOP shopping centers (Bologna, Modena, Genoa, Rimini, and Grosseto, May–June 2017), Mese del Cuore (Rome, September–October 2017), Tennis and Friends (Rome, October 2017), and CONAD shopping centers (Terni, Perugia, Viterbo, Anzio, Caserta, November 2017). For the present study, 133 participants were excluded for missing values in the variables of interest; as a consequence, a sample of 7907 participants was considered.

### 2.2. Data Collection

All participants who accepted to participate the Lookup 7+ activity received a dedicated visit. The Lookup 7+ check-up is designed to obtain the following data: informed consent, age, gender, lifestyle habits (smoking, eating habits, and physical activity), blood pressure measurement, weight and height assessment, cholesterol measurement, fasting glucose measurement, and physical performance test [24,25].

### 2.3. Anthropometric Evaluation

Well-trained examiners, measured anthropometric indices with participants wearing light, thin clothing and no shoes. Body weight was measured through an analogue medical scale. Body height was measured using a standard stadiometer. Body weight and height were measured to the nearest 0.1 kg and 0.1 cm, respectively.

BMI was defined as weight (kilograms) divided by the square of height (meters). The BMI cutoffs recommended by the WHO were used. The classes of BMI reported by the WHO are: 18.5–24.9 kg/m^2^ for normal, 25.0–29.9 kg/m^2^ for overweight, and > 30 kg/m^2^, for obesity. Obesity was categorized as class I (30–34.00 kg/m^2^), class II (35–39.99 kg/m^2^), and class III (> 40 kg/m^2^) (www.who.int/nutrition/publications/obesity/WHO_TRS_894).

### 2.4. Blood Pressure Measurement

Blood pressure was measured—in the sample of Milan EXPO 2015—with a clinically validated Omron M6 electronic sphygmomanometer (Omron, Kyoto, Japan) and in all other samples with a manual sphygmomanometer according to recommendations from international guidelines [26]. In all participants, blood pressure measurements were obtained after resting for 5 min in a seated position, with 30 s intervals between cuff inflations. An average of three measurements were used. Care was taken to select the cuff size according to the participant’s arm circumference. In all the experimental settings, the assessment was performed in a dedicated room, with an optimal room temperature, and respecting the privacy.

Blood pressure values were categorized as lower than 120/80 mmHg (if untreated), between 120/80 and 139/89 mmHg (or treated to goal), and equal to or higher than 140/90 mmHg. Antihypertensive drug use was also recorded. Hypertension was defined as mean systolic blood pressure ≥ 140 mmHg and/or diastolic blood pressure ≥ 90 mmHg and/or anti-hypertensive drug use [27].

### 2.5. Lifestyle Assessment

Smoking status was categorized as current or never/former smoker [25]. Healthy diet was considered as the consumption of at least three portions of fruit and/or vegetables per day [28]. Daily intake of fruit and vegetables was calculated on the reference tables for the Italian population released by the Italian Society of Nutrition (SINU). Accordingly, three or more portions of fruit and/or vegetables correspond approximately to 400 g, which is the minimum recommended by the WHO. The use of three or more portions to identify a healthy diet is in line with Italian dietary habits for fruit and vegetables, which are typically eaten during the main meals rather than as snacks. Reference amounts are available at http://www.sinu.it/html/cnt/larn.asp.

Regular participation in physical activity was considered as involvement in exercise training at least twice a week during the last year. Accordingly, participants were considered physically active or inactive. To be assigned in the active group, the following activities were considered: walking for at least 30 min per session, cycling, swimming, running, and resistance exercise [20,29].

### 2.6. Blood Measurements

Cholesterol was measured from capillary blood samples using disposable electrode strips based on a reflectometric system with a portable device (MultiCare-In, Biomedical Systems International Srl, Florence, Italy) [30]. The inter-assay imprecision (expressed as variation coefficient) of the MultiCare system was 4.51% (range, 2.38–8.54%); sensitivity and specificity measurements were 95.7% and 61.9% (threshold value of cholesterol 190 mg/dL). Random blood glucose was measured from capillary blood samples using disposable electrode strips based on an amperometric system with a MultiCare-In portable device [30]. Participants who declared being diabetic and those who presented with a random blood glucose level > 200 mg/dL were considered to be suffering from diabetes.

### 2.7. Statistical Analyses

Continuous variables are expressed as mean ± standard deviation (SD), categorical variables as frequencies by absolute value and percentage (%) of the total. Descriptive statistics were used to define demographic and key clinical characteristics of the study population according to the presence of hypertension. Differences in proportions and the means of covariates were evaluated using Fisher’s Exact Test and *t* test statistics, respectively.

Logistic regression analysis was used to assess the association between different levels of BMI (normal weight, overweight, and obesity) and hypertension. To identify factors independently associated with the presence of hypertension, we first estimated a crude prevalence rate ratio (PRR) at a 95% confidence interval (CI) and then controlled for age and gender (model 1). A logistic regression analysis was computed including the five cardiovascular health metrics (smoking, healthy diet, physical activity, cholesterol, and serum glucose level).

Finally, analysis of covariance (ANCOVA)—adjusted for age—was used to examine the effect of BMI levels on systolic and diastolic blood pressure.

All analyses were performed using SPSS software (version 11.0, SPSS Inc., Chicago, IL, USA).

## 3. Results

Mean age of 7907 volunteers participating in the Longevity check-7+ surveys was 55.4 (SD: 15.0, range: 18–98) years, and 4480 (56%) were women. Characteristics of the study population according to the presence of hypertension are summarized in Table 1. As compared with women, men had a higher prevalence of hypertension. Participants with hypertension were significantly older than those without hypertension (64.4 years versus 48.5 years, respectively; *p* < 0.001). In particular, hypertensive participants showed higher prevalence of diabetes and cholesterol levels than non-hypertensive enrollees. Finally, the BMI value was significantly higher in participants with hypertension than in non-hypertensive participants (26.7 Kg/m^2^ versus 24.1 Kg/m^2^, respectively; *p* < 0.001).

Age-adjusted results from ANCOVA models showing the differences of diastolic and systolic blood pressure across different BMI levels are reported in Figure 1. After adjusting for age, significantly different distributions of systolic and diastolic blood pressure across BMI levels were consistent. Overall, the average systolic blood pressure (Figure 1, Panel A) and diastolic blood pressure (Figure 1, Panel B) increased significantly and linearly across BMI levels. The systolic blood pressure increased by more than 10 mmHg from a normal BMI to a BMI above 40 Kg/m^2^ (class III obesity) (123 mmHg versus 135 mmHg, respectively; *p* < 0.001), in both men and women (*p* for trend < 0.001). Similarly, the diastolic blood pressure increased by more than 5 mmHg from a normal BMI to a BMI above 40 Kg/m^2^ (class III obesity) (74 mmHg versus 81 mmHg, respectively; *p* < 0.001), in both men and women (*p* for trend <0.001).

According to the BMI cutoffs recommended by the WHO, overweight status was observed among 2896 (38%) participants; the obesity status was identified in 1135 participants (15%), with 893 (11.8%) participants in class I, 186 (2.5%) in class II, and 56 (0.7%) in class III. As shown in Figure 2, the prevalence of hypertension significantly increased with the increase in BMI. Among participants with a normal BMI, the prevalence of hypertension was 45% compared to 67% among overweight participants, 79% in obesity class I and II, and up to 87% among participants with obesity class III (*p* for trend < 0.001).

In the unadjusted model, there was a direct association between BMI levels and hypertension, starting from overweight [odds ratio (OR) 2.49, 95% CI 1.51–2.78] to class III obesity (OR 8.58, 95% CI 3.87–19.00) (Table 2). After adjusting for potential confounders—age, gender, smoking habit, healthy diet, physical activity, cholesterol and glucose levels—this association remained statistically significant. In the fully adjusted model, participants with class III obesity had a significantly increased risk of hypertension compared with those with normal BMI (OR 6.49, 95% CI 2.85–14.78) (Table 2, Model 2).

We also tested the possible interaction between gender and BMI for the diagnosis of hypertension, but no significant result was reported. Finally, no interaction between season’s assessment and blood pressure was observed.

## 4. Discussion

The Lookup 7+ project evaluated established cardiovascular health metrics in a large and unselected cohort of Italian community-dwelling men and women ranging between 18 and 98 years. In this respect, the Lookup 7+ project provided the unique opportunity of assessing these domains among people of both genders across a wide age range outside of conventional healthcare or research settings [22]. Using this innovative database, the present study shows that both systolic and diastolic blood pressure values are linearly correlated with BMI; in particular, overweight and obesity status were significant risk factors for hypertension. While age and gender usually affect the prevalence of hypertension, BMI remained the main determinant when the analyses were stratified by gender and adjusted for age. A significant association was also observed for BMI even after adjusting for other covariates, suggesting that overweight and obesity *per se* may lead to the development of hypertension and play a central role in its pathogenesis [31].

The overweight status, which reflects increased body fat mass, was demonstrated to be an independent risk factor for hypertension, which was consistent with previous studies showing an association between high body fat levels and hypertension [7,10,32,33,34]. However, the exact mechanism underlying the association of visceral fat and hypertension remains unknown. Inflammatory processes have been shown to play an important role in the mechanisms involved in the pathogenesis of hypertension [35]. Fat cells are characterized by being sensitive to lipolysis and by their aptitude to produce high quantities of inflammatory cytokines. This inflammatory response participates in blood pressure elevation and end-organ damage. Furthermore, it is possible that increased adipose tissue releases a variety of adipokines that are related to a decrease in the production and use of nitric oxide, which has important functions in the control of vascular tone and suppression of vascular smooth muscle cell proliferation. A decrease in the effect of nitric oxide has been associated with endothelial dysfunction and arterial hypertension [36].

Overall, the cardiovascular health metrics score has recently been associated with a lower risk of cardiovascular and non-cardiovascular mortality, with clear benefits on arterial stiffness [37], carotid intima media thickness [38], and coronary artery calcification [39], linking the potential importance of these factors to improvements in health and successful longevity free of CVD. In particular, vascular risk factors, which are represented by ideal cardiovascular health metrics components and, in particular, BMI may be correlated not only with clear signs and symptoms of diseases, but also with an accumulation of subclinical vascular disorders, resulting in a higher blood pressure preceding the onset of clinically evident manifestation of hypertension [40].

Our results highlight the importance of adopting the cardiovascular health metrics as a “primordial” prevention strategy [41]. In this respect, the American Heart Association’s 2020 Strategic Impact Goals are set to reduce the burden of CVD by 20% and increase CVH by 20% by the year 202 [42]. Along with prior publications, the present study demonstrates the importance of promoting ideal and normal body weight as a national strategy, not just for the general reduction of CVD, but also for its favorable impact on blood pressure. As a consequence, it is reasonable to expect a parallel decrease in hypertension and its negative consequences and an improvement in population health and wellbeing beyond the targeted 20% reduction of CVD.

Body weight and BMI are easily measured and are simple and effective tools for screening the risk of hypertension, making these anthropometric measures suitable for use in comprehensive public health strategies. The prevalence of obesity has been increasing significantly in Western countries over recent years, and the burden of hypertension is expected to continue to increase. Therefore, the use of BMI should be recommended when looking to predict and screen for hypertension.

Albeit dealing with a highly relevant issue, our study presents some important limitations that need to be discussed. First, the results shown in this paper were obtained from a cross-sectional survey limiting the ability to draw cause-and-effect implications between different levels of BMI and hypertension. As a consequence the ability to recognize whether a normal BMI is correlated with the postponement of higher blood pressure over time is limited. Second, the type of evaluation could influence the assessment of some health metrics. For example, the chosen setting of Milan EXPO 2015 or some shopping centers could lead to an overestimation of the blood pressure. Even though the blood pressure was measured according to recommendations from international guidelines, people who decided to participate in the study procedures were involved—before being assessed—in usual activities, such as walking, carrying bags, and eating. The activities, performed immediately before being evaluated, could have influenced the assessment. Limitations also include lack of information about conditions, such as stroke, myocardial infarct, and other CVD that have a direct impact on blood pressure. However, for the type of participants recruited in the study, it is possible to exclude that acute illnesses were present at the time of evaluation. Furthermore, given the type of the check-up, we have no information about triglycerides, HDL and LDL cholesterol levels. Only in the last manifestations, the waist and hip circumferences has been added to the Lookup 7+ assessment. For this reason, it was not possible to take this important parameter into consideration. A deeper understanding of the relationship between BMI and hypertension requires the analysis of prospective data that are not available at this stage for our study. Finally, the population included only Caucasian persons, so our results may not be applicable to other ethnic groups.

Apart from these limitations, this study offered a unique opportunity to investigate the impact of BMI on blood pressure. We found a gradient of increasing blood pressure with higher levels of BMI. The fact that this gradient is present even in the fully adjusted analyses suggests that BMI may cause a direct effect on blood pressure, independent of other clinical risk factors. Nevertheless, despite extensive research efforts, the mechanism responsible for BMI-associated improvement in blood pressure has not been completely elucidated. For example, the connections among obesity, diabetes, and hypertension could explain, at least in part, the results observed. Diabetes represents an independent risk factor for CVD and at the same time obesity is a risk factor for diabetes. This vicious association observed in overweight and obese people can be interrupted by an adequate control of body weight [43]. 

On the basis of our results, it is possible to confirm that BMI was an independent risk factor for hypertension. Therefore, BMI measurement should be recommended as a simple and effective predictor of hypertension in public health strategies [44,45]. Furthermore, many of the strategies that produce successful weight loss and maintenance will help prevent obesity. For example, improving eating habits and increasing physical activity play a vital role in controlling BMI, and as a consequence, in reducing the risk of adverse health outcomes.

Finally, body fat distribution is also an important risk factor for obesity-related diseases. Abdominal fat is related to an increased risk of CVD. In clinical practice, waist circumference is used as a surrogate marker of abdominal fat mass, because it correlates with abdominal fat mass and is associated with cardio-metabolic disease risk [46]. The relative gender-specific cut-points were derived from a regression curve that identified the waist circumference values associated with a BMI ≥ 30 kg/m^2^ in primarily Caucasian men and women and are unlikely to affect clinical management when BMI and other obesity-related cardio-metabolic risk factors are already being determined [46].

## Figures and Tables

**Figure 1 nutrients-10-01976-f001:**
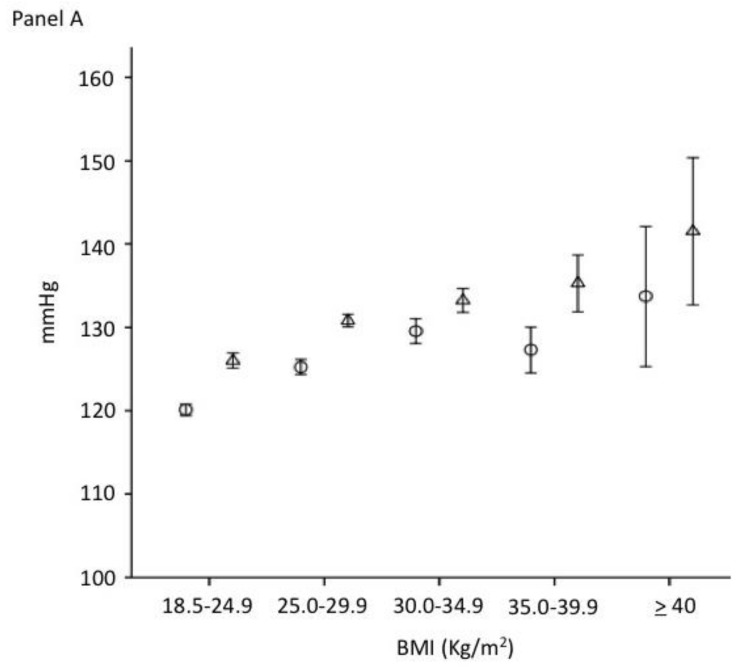
Systolic (panel **A**) and diastolic (panel **B**) blood pressure (mean and SD) according to the BMI levels (○: women; △: men) (*p* for trend < 0.001).

**Figure 2 nutrients-10-01976-f002:**
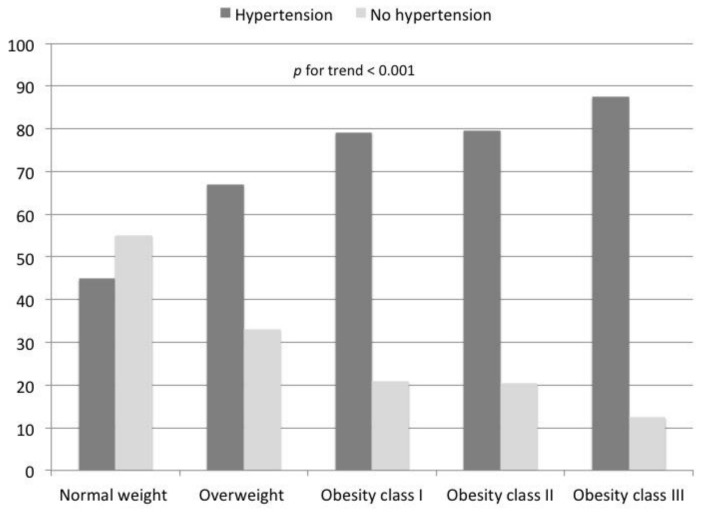
Prevalence of hypertension according to different levels of BMI.

**Table 1 nutrients-10-01976-t001:** Characteristics of the study population according to hypertension *.

Characteristics	Total Sample	Hypertension	No Hypertension	
	(*n* = 7907)	(*n* = 4588)	(*n* = 3319)	*p*
Age (years)	55.4 ± 15.0	60.4 ± 13.5	48.5 ± 14.2	<0.001
Gender				
Male	3427 (44)	2309 (67)	1116 (33)	<0.001
Female	4480 (56)	2279 (51)	2203 (49)	
Smoking	1348 (17)	648 (14)	700 (21)	<0.01
Healthy diet	5586 (71)	3346 (72)	2240 (68)	<0.01
Physically active	4337 (55)	2498 (54)	1839 (56)	0.22
BMI (Kg/m^2^)	25.6 ± 4.3	26.7 ± 4.3	24.1 ± 3.8	<0.001
Cholesterol (mg/dL)	210.5 ± 34.1	211.3 ± 32.7	209.9 ± 35.1	0.09
Diabetes	639 (8)	498 (11)	141 (4)	<0.001
Systolic blood pressure (mmHg)	125.4 ± 16.9	135.2 ± 14.5	112.4 ± 9.4	<0.001
Diastolic blood pressure (mmHg)	75.7 ± 10.1	79.6 ± 9.6	70.4 ± 7.8	<0.001

* Data are given as number (percent) for gender, smoking, and healthy diet, physical activity, diabetes; for all the other variables, means ± SD are reported. Healthy diet: consumption of at least three portions of fruit and/or vegetables per day. Physically active: physical exercise at least twice a week. BMI: body mass index.

**Table 2 nutrients-10-01976-t002:** Unadjusted and adjusted association (OR and 95% CI) between body mass index and hypertension.

Body Mass Index (BMI)	Univariate Odds Ratio (95% CI)	Adjusted Odds Ratio Model 1 (95% CI)	Adjusted Odds Ratio Model 2 (95% CI)
Normal weight (18.5–24.9 Kg/m^2^)	Referent	Referent	Referent
Overweight (25.0–29.9 Kg/m^2^)	2.49 (1.51–2.76)	1.73 (1.55–1.94)	1.73 (1.54–1.95)
Class I obesity (30.0–34.9 Kg/m^2^)	4.66 (3.91–5.55)	3.24 (2.69–3.91)	3.38 (2.79–4.10)
Class II obesity (35.0–39.9 Kg/m^2^)	4.77 (3.32–6.86)	4.27 (2.90–6.27)	4.62 (3.08–6.93)
Class III obesity (> 40 Kg/m^2^)	8.58 (3.87–19.00)	6.51 (2.88–14.69)	6.53 (2.87–14.85)

Model 1: adjusted for age and gender; Model 2: adjusted for age, gender, smoking habit, healthy diet, physical activity, cholesterol, and diabetes.

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
