# Peer review of "Body Mass Index is Strongly Associated with Hypertension: Results from the Longevity Check-Up 7+ Study"

_nutrients, 2018, doi:10.3390/nu10121976_

Round 1
Reviewer 1 Report
The study titled “Body Mass Index is Strongly Associated with Hypertension: Results from Longevity Check-up 7+ Study” by Francesco et al.
There are some minor concerns to be addressed.
1. What is the accuracy level of Cholesterol testing strips based test.
2. Does this study has any significant gender difference with aging in BMI associated Hypertension.
3. Does authors tested triglycerides, HDL and LDL levels? Because these are key components in BMI and can influence the development of Cardio Vascular Diseases.
4. Authors should include preventable measures for BMI associated hypertension in the discussion.
Author Response
1. What is the accuracy level of Cholesterol testing strips based test.
We sincerely thank the reviewer for this and the following suggestions. We added this information in revised version of paper. The inter-assay imprecision (expressed as variation coefficient) of the MultiCare system was 4.51% (range, 2.38%-8.54%); sensitivity and specificity measurements were 95.7% and 61.9% (threshold value of cholesterol 190 mg/dL). Accordingly, the text has been modified and new sentence has been added in the methods section.
(Page 4, 1st paragraph)
2. Does this study have any significant gender difference with aging in BMI associated Hypertension?
We tested the possible interaction between gender and BMI for the diagnosis of hypertension, but no significant result was reported. This issue is now addressed in the results section. The text has been modified and a new sentence has been added.
(Page 7, 2nd paragraph)
3. Does authors tested triglycerides, HDL and LDL levels? Because these are key components in BMI and can influence the development of Cardio Vascular Diseases.
Given the type of the check-up, we have no information about triglycerides, HDL and LDL cholesterol levels. This issue is now addressed as a potential limitation of the study. New sentence has been added in the discussion section.
(Page 8, 4th paragraph)
4. Authors should include preventable measures for BMI associated hypertension in the discussion.
We thank the reviewer for arising this issue. At the end of the discussion section we add two new sentences to address this point. Accordingly the text has been modified.
(Page 9, 1st paragraph)
Reviewer 2 Report
Please see my comments below:
1) Why there are some texts highlighted in yellow or green?
2) Table 1: What is the definition of healthy diet? This can vary significantly across countries. The current defintion is too general. The authors should look into the types of diet, for example, Mediterranean diet.
3) Methods: How about the assessment of waist circumference? The fact thatBMI is associated with a higher systolic blood pressure is not something new. The novelty of this study is its sample size.
4) The authors should include the asessment of waist circumference because this is a better measurement than BMI. BMI has been reported to be not an accurate measurement and waist circumference has been suggested to perform better than BMI because waist circumference takes into account the fat mass.
5) For the blood pressure measurement, please describe the setting of the measurement taking place. Also, please check for the confounders such as seasons, because winter time, the blood pressure tends to be elevated. Were these confounders taken into consideration?
6) Please also give a more detailed physically active definition. The current definition is too broad.
7) Was fasting glucose taken for measurement?
8) How about the presence of other chronic diseases such as CVD or diabetes in participants? Were these variables collected?
9) Overall, the authors were trying to use big data to answer the research question. However, the research question was already being answered by several larger intervention studies and meta analyses. Therefore, the authors should have include some other measurements such as waist circumference. Otherwise, this study does not appear to contribute any new findings except its big sample size.
10) please check the reference style. It appeared that the authors did not pay attention to this detail.
Author Response
1) Why there are some texts highlighted in yellow or green?
The editorial office for internal use edited the previous texts highlighted in yellow or green. In this revised version of the paper the text highlighted in yellow indicates the correction made.
2) Table 1: What is the definition of healthy diet? This can vary significantly across countries. The current definition is too general. The authors should look into the types of diet, for example, Mediterranean diet.
We sincerely thank the reviewer for this and the following suggestions.
The definition of healthy diet has been better clarified. Daily intake of fruit and vegetables was calculated on the reference tables for the Italian population released by the Italian Society of Nutrition (SINU). Three or more portions of fruit and/or vegetables correspond approximately to 400g, which is the minimum recommended by the World Health Organization. The use of three or more portions to identify a healthy diet is in line with Italian dietary habits for fruit and vegetables, which are typically eaten during the main meals rather than as snacks. Accordingly the text has been modified and new sentences have been added in the methods section.
(Page 3, 7th paragraph)
3) Methods: How about the assessment of waist circumference? The fact that BMI is associated with a higher systolic blood pressure is not something new. The novelty of this study is its sample size.
Only in the last manifestation the waist and hip circumferences have been added to the Look-up 7+ assessment (few subjects already evaluated). For this reason it was not possible to take this important parameter into consideration. This issue has been addressed as a limitation of study.
However, it is important to highlight that in clinical practice waist circumference is used as a surrogate marker of abdominal fat mass, because it correlates with abdominal fat mass and is associated with cardio-metabolic disease risk. The relative gender specific cut points were derived from a regression curve that identified the waist circumference values associated with a BMI ≥30 kg/m2 in primarily Caucasian men and women and are unlikely to affect clinical management when BMI and other obesity-related cardio-metabolic risk factors are already being determined. This issue has been addressed in the discussion section. Accordingly the text has been modified and a new sentence has been quoted.
(Page 8, 4th paragraph; Page 9, 2nd paragraph; Reference number 45)
4) The authors should include the assessment of waist circumference because this is a better measurement than BMI. BMI has been reported to be not an accurate measurement and waist circumference has been suggested to perform better than BMI because waist circumference takes into account the fat mass.
This issue has been addressed as a limitation of study. See answer number 3, too.
(Page 8, 4th paragraph)
5) For the blood pressure measurement, please describe the setting of the measurement-taking place. Also, please check for the confounders such as seasons, because wintertime, the blood pressure tends to be elevated. Were these confounders taken into consideration?
In all the experimental settings the assessment was performed in a dedicated room, with an optimal room temperature, and respecting the privacy. Furthermore, no interaction between season’s assessment and blood pressure was observed. Accordingly the text has been modified and new sentences have been added in the methods and results sections.
(Page 3, 4th paragraph; Page 7, 2nd paragraph)
6) Please also give a more detailed physically active definition. The current definition is too broad.
We have better specify the time considered in the assessment. Regular participation in physical activity was considered as involvement in exercise training at least twice a week during the last year. Otherwise, the physically active definition has been already addressed in the methods section, quoting two specific references, too.
(Page 7, 8th paragraph)
7) Was fasting glucose taken for measurement?
Random blood glucose was measured from capillary blood samples. This issue has been now clarified in the methods section.
(Page 4, 1st paragraph)
8) How about the presence of other chronic diseases such as CVD or diabetes in participants? Were these variables collected?
We sincerely thank the reviewer to arise this issue. Regarding the cardiovascular diseases, we have no this information. However, this issue has been already addressed as a potential limitation of the study. “Limitations also include lack of information about conditions, such as stroke, myocardial infarct, and other cardiovascular diseases that have a direct impact on blood pressure. However, it is important to highlight that for the type of participants recruited in the study sample, it is possible to exclude that acute illnesses were present at the time of evaluation”.
(Page 8, 4th paragraph)
Regarding the presence of diabetes we have this information, because it is part of the CV health metrics assessment. We have clarified in the methods section how we assessed blood glucose and diabetes in our sample. In the revised version of the manuscript we now consider diabetes and no the absolute blood glucose levels. Accordingly, the text has been modified and the new data are presented in Table 1 and Table 2.
(Page 4, 5th paragraph; Tables 1 and 2)
9) The authors should have included some other measurements such as waist circumference.
This issue has been addressed as a limitation of study. See answers number 3 and 4, too.
(Page 8, 4th paragraph)
10) Please check the reference style. It appeared that the authors did not pay attention to this detail.
We have revised the reference style where possible. If the Editor needs further editing we are ready to do it.
Round 2
Reviewer 2 Report
I have only a couple of minor comments:
1) Are the format of in-text citations correct according to the journal format?
2) Please use "participants" instead of "subjects" throughout the text.
Author Response
1) Are the format of in-text citations correct according to the journal format?
We have modified the format in the text.
2) Please use "participants" instead of "subjects" throughout the text.
Accordingly, the text has been modified.